# Nonintrusive thermal-wave sensor for operando quantification of degradation in commercial batteries

Yuqiang Zeng [1,2], Fengyu Shen[2], Buyi Zhang [2,3], Jaeheon Lee [2,4], Divya Chalise [2,3], Qiye Zheng[2,5], Yanbao Fu[2], Sumanjeet Kaur [2], Sean D. Lubner[2], Vincent S. Battaglia[2], Bryan D. McCloskey [2,4], Michael C. Tucker [2] & Ravi S. Prasher [2,3] ✉

Monitoring real-world battery degradation is crucial for the widespread application of batteries in different scenarios. However, acquiring quantitative degradation information in operating commercial cells is challenging due to the complex, embedded, and/or qualitative nature of most existing sensing techniques. This process is essentially limited by the type of signals used for detection. Here, we report the use of effective battery thermal conductivity ($k_{eff}$) as a quantitative indicator of battery degradation by leveraging the strong dependence of $k_{eff}$ on battery-structure changes. A measurement scheme based on attachable thermal-wave sensors is developed for non-embedded detection and quantitative assessment. A proof-of-concept study of battery degradation during fast charging demonstrates that the amount of lithium plating and electrolyte consumption associated with the side reactions on the graphite anode and deposited lithium can be quantitatively distinguished using our method. Therefore, this work opens the door to the quantitative evaluation of battery degradation using simple non-embedded thermal-wave sensors.

Rechargeable batteries play an essential role in the ongoing development of renewable energy[1–3]. Battery operation under certain conditions (e.g., extreme temperatures[4] and/or rates[5]) can cause early degradation and thermal safety issues. Probing and understanding the real-world degradation are key to the improvement of battery lifetime, safety, and reliability in practical applications. The two main origins of battery degradation are lithium plating and solid-electrolyte interphase (SEI) growth, which lead to aging phenomena such as the loss of lithium-ion inventory[6] and electrolyte dry out[7]. For example, lithium plating dominates the capacity fade during battery operation at high rates and/or low temperatures, while high operation temperature accelerates the growth of SEI and the consumption of electrolyte,

leading to rapid capacity loss. Various sensing techniques using different signals (temperature[8,9]; pressure[10]; electrochemical[11,12], acoustic[13–16], and optical signals[17–20]) have been developed to monitor the internal change and aging of batteries. Among these methods, acoustic and optical sensing techniques have received the most attention because of their capability to detect various types of degradation[21].

Acoustic sensing is a highly sensitive nonintrusive technique that relies on the propagation velocity and amplitude attenuation of acoustic waves across the battery. These features are beneficial for investigating a specific effect in controlled experiments, e.g., electrolyte wetting and drying[15]. However, the high sensitivity to many

[1]School of Microelectronics, Southern University of Science and Technology, Shenzhen 518055, China. [2]Energy Storage and Distributed Resources Division, Lawrence Berkeley National Laboratory, Berkeley, CA 94720, USA. [3]Department of Mechanical Engineering, University of California, Berkeley, CA 94720, USA. [4]Department of Chemical and Biomolecular Engineering, University of California, Berkeley, CA 94720, USA. [5]Mechanical and Aerospace Engineering Department, The Hong Kong University of Science and Technology, Hong Kong, China. ✉e-mail: rsprasher@lbl.gov

coupled physical–thermal–chemical parameters is unfavorable for distinguishing and quantifying the exact sources in commercial systems. Recently, optical sensors have been used to decipher the conjoined information, e.g., temperature and pressure can be decoupled using multiple sensors with different sensitivities to these parameters[18]. Optical sensing provides clear physical, thermal, and chemical information for battery R&D that was previously inaccessible. Nevertheless, the embedded nature of optical sensors (i.e., the preparation of customized cells) raises issues for their use in commercial batteries[21]. The sensor lifetime is far below the cycle life of commercial batteries due to the poor chemical stability of optical fibers in a harsh corrosive electrochemical environment. The other major concern regarding embedded sensors is their incompatibility with existing battery manufacturing technique and the additional manufacturing cost[4,21]. Thus, neither of these sensing techniques can provide long-term monitoring of battery degradation or obtain quantitative chemical information for commercial cells in complex practical scenarios.

To monitor the degradation in real-world systems, an ideal sensing technique should be nonintrusive (i.e., non-embedded) and the signal should be selectively sensitive to the key parameters related to various types of battery degradation[6,7] such as lithium plating, electrolyte dry out, and loss of active material. Recently, we linked the amount of intercalated lithium ions to the electrode thermal conductivity, which led to the first-time use of embedded thermal-conductivity measurement for spatial mapping of lithium concentration across battery electrodes[22]. In this work, we demonstrate a non-embedded thermal-wave sensing technique (also known as 3 omega sensors[22]) for accurately tracking the evolution of various degradation sources from the measured effective battery thermal conductivity ($k_{eff}$). We developed a measurement scheme to calibrate and leverage the quantitative relationship between $k_{eff}$ and battery degradation (e.g., Li plating and electrolyte consumption). Simple attachable thermal-wave sensors were fabricated for the $k_{eff}$ measurement, leading to completely non-embedded detection. Controlled

experiments and in-situ characterization using X-ray tomography were performed to validate our approach. Further, a case study of commercial lithium-ion batteries (LIBs) during fast charging demonstrates the use of our technique in quantitatively distinguishing the degradation sources.

## Results and discussion
### Thermal conductivity of lithium-ion batteries

A unit cell of a battery consists of current collectors, a porous separator, and electrodes (Fig. 1a). The total thermal impedance of a unit cell is $R_{tot} = \sum_{i=1}^{5} \frac{L_i}{k_{L,i}} + TCR_{sep-a} + TCR_{sep-c}$, where $L_i$, $k_{L,i}$, $TCR_{sep-a}$, and $TCR_{sep-c}$ are the thickness and thermal conductivity of the $i_{th}$ layer and the thermal contact resistance ($TCR$) between the separator and electrodes (anode and cathode), respectively. The effective battery thermal conductivity depends on both the layer and interface properties and is given by

$$k_{eff} = \sum_{i=1}^{5} L_i / R_{tot}. \tag{1}$$

During battery operation, the components in $k_{eff}$ that vary as a function of time are 1) the thermal conductivity of the electrodes and separator ($k_{L,i}$) and 2) $TCR$.

Thermal conductivity of the layers: The thermal conductivities of the collectors are known as they are composed of Al and Cu. For the porous separator and electrode layers, the thermal conductivity is a function of the bulk porosity and the thermal conductivity of the solid material ($k_s$) and fluid part ($k_f$). The $k_s$ is either known from the literature or can be obtained by measuring the thermal conductivity of the electrolyte-wetted or dry electrodes and applying effective medium theory (e.g., the Bruggeman model[23]), as described in Methods. As the changes of the porosity and $k_s$ are negligible compared to the change of $k_f$, the decrease in the thermal conductivity of the porous layers is

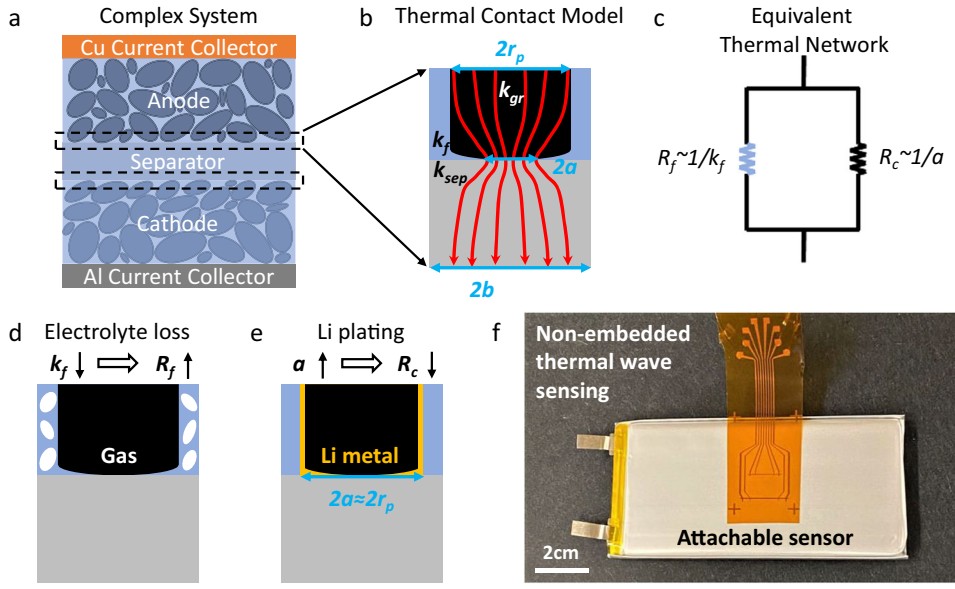

**Fig. 1 | Nonintrusive monitoring of battery degradation using non-embedded thermal-wave sensors. a** Schematic of a unit cell of complex porous electrodes. **b** Thermal model for the contact between the separator and electrodes describing the role of the particle radius ($r_p$), constriction radius ($a$), half width of the unit cell ($b$), and the thermal conductivity of fluid ($k_f$), graphite particle ($k_{gr}$), and separator ($k_{sep}$). **c** Equivalent thermal network for the contact: the constriction resistance ($R_c$) and fluidic resistance ($R_f$). This model quantifies the effect of (**d**) electrolyte

consumption and (**e**) Li plating on the thermal contact resistance. It is noteworthy that the drop in $k_f$ due to electrolyte consumption results in an increase of both the thermal contact resistance ($TCR$) and thermal resistance of the porous layers, whereas the effective increase of $a$ due to Li plating only reduces $TCR$. **f**, Attachable thermal-wave sensors for monitoring the evolution of $k_{eff}$ and corresponding degradation sources.

dominated by the drop of $k_f$ due to electrolyte dry out (mixture of liquid and gas), i.e., 0.23 W/m·K for pure electrolyte and 0.025 W/m·K for pure gas.

The thermal contact resistance between the separator and electrodes (*TCR*): Physics and the model for *TCR* between the solid surface and particles, with interstitial fluid, is very well described in many previous works[24–26]. *TCR* depends on the thermal conductivity of the solid particles, fluid, and solid substrate as well as the constriction radius (*a*), as shown in Fig. 1b. Figure 1c shows the thermal-resistance network near the interface. The constriction radius depends on the particle size of the electrode materials such as NMC or graphite. Because the particle size in real batteries has a wide range (Fig. 1a), the constriction radius will also vary accordingly. For simplicity, an average constriction radius is used in our thermal model. Details on the calculation of the constriction resistance ($R_c$) can be found in Methods. Note that $R_c$ varies inversely with the constriction radius ($1/a$). The fluidic resistance near the interface depends on surface porosity, $1 - \pi r_p^2 / 4b^2$, where $b$ represents the average size of an equivalent unit cell near the interface (Fig. 1b). The corresponding thermal resistance ($R_f$) can be given as $R_f = \frac{r_p}{k_f(4b^2 - \pi r_p^2)}$, where $r_p$ is the average electrode particle radius (available from the supplier or can be measured). The two parameters *a* and *b* are obtained from a calibration experiment. From the known *a* and *b*, the *TCR* can be calculated as $TCR = 4b^2/(1/R_c + 1/R_f)$.

From the thermal model, the effective $k_f$ decreases with electrolyte consumption, which will increase *TCR* and the thermal resistance of the porous layers (Fig. 1d) and thus decrease $k_{eff}$ (Eq. (1)). In contrast, the deposition of thermally conductive lithium metal (~85 W/m·K) on anode particles can be approximated as high-thermal-conductivity fillers between the anode and separator (Fig. 1e), which effectively increases the constriction radius *a* and reduces *TCR*, causing an increase of $k_{eff}$. Besides, cycling induced cathode cracking may result in loss of contact inside the cathode particles, and thus increases thermal constriction resistance and degrades the interfacial thermal transport. However, this effect proved to be weak in our case studies as discussed later. We speculate that the impact of cathode change could be significant in certain extreme conditions (e.g., severe pulverization) and should be studied in the future. These opposing trends present an opportunity to quantitatively distinguish the degradation mechanisms (e.g., Li plating and electrolyte consumption) via thermal-conductivity measurement.

## Calibration and validation of the thermal model

The two fit parameters, *a* and *b*, were obtained through calibration experiments, which could be obtained from embedded or non-embedded $k_{eff}$ measurement. For both calibration and validation, analogous to our previous work[22], we prepared batteries of single unit cells using NMC/Gr electrodes (see Supplementary Table 1 for material properties and Supplementary Table 2 for a summary of the thermal properties) with embedded thermal sensors. The sensor fabrication (see Supplementary Fig. 1) and thermal-conductivity measurement procedure has been detailed in Methods. The calibration consists of measuring $k_{eff}$ in fully dry and wet conditions, where $k_f$ is known (0.025 W/m·K for gas and 0.23 W/m·K for the electrolyte), and fitting the measured $k_{eff}$ vs. $k_f$ to Eq. (1). For the battery considered in this study, these two fit parameters were determined to be $a/r_p = 0.28$ and $b/r_p = 1.18$ (see raw data and representative fit in Supplementary Fig. 2).

The robustness of the thermal model (Eq. 1) with parameters *a* and *b* obtained from the calibration experiments was evaluated by comparing the measured and calculated $k_{eff}$ associated with different fluid thermal conductivity and lithium coverage. First, we measured $k_{eff}$ when the cell was completely wetted using other fluids of different $k_f$, e.g., isopropanol (IPA) with $k_f = 0.14$ W/m·K and a mixture

of IPA:H$_2$O = 1:1 with $k_f = 0.35$ W/m·K. Figure 2a shows the good agreement between the measured $k_{eff}$ and the $k_{eff}$ calculated using Eq. 1 and a and b obtained from the calibration experiments for a broad range of $k_f$, which proves the accuracy of our thermal model for different $k_f$. In real LIBs, the calibrated model predicts that $k_{eff}$ decreases from ~0.4 to ~0.2 W/m·K as $k_f$ decreases due to electrolyte consumption (Supplementary Fig. 3). As $k_{eff}$ can be measured from experiments, the effective fluid conductivity ($k_{f,eff}$) for the corresponding $k_{eff}$ can be back calculated using our thermal model (Supplementary Fig. 3). Once $k_{f,eff}$ is known, the amount of electrolyte consumption ($\phi_{dry}$) can be calculated. Because the fluid is a mixture of liquid and gas bubbles when electrolyte dry out occurs, composite mixing model (Bruggeman model, see Methods) can be used to extract $\phi_{dry}$ by fitting the model to $k_{f,eff}$.

In addition, we verified the accuracy of the thermal model in assessing the fraction of anode particles that are covered with Li metal ($\phi_{Li}$), as Li plating does not occur uniformly (Fig. 2b). During battery aging, we assume that *b* will remain the same as the variation of electrode area with aging is negligible, whereas only *a* will change with Li plating. For a surface anode particle covered with Li metal, its constriction radius becomes the same as the particle radius ($a = r_p$), as shown in Fig. 1e. For partial Li deposition coverage with $\phi_{Li} < 100\%$ (Fig. 2b), only the surface anode particles covered with Li metal have $a = r_p$, whereas the surface particles without Li plating maintain the same constriction radius, as determined from calibration experiments. Because the surface anode particles have two different constriction resistances due to partial Li coverage, the thermal resistance of the unit cell can be split into two parallel paths weighted by $\phi_{Li}$, i.e., $R_{tot}^{-1} = \phi_{Li} R_{tot,Li}^{-1} + (1 - \phi_{Li}) R_{tot,0}^{-1}$, and the effective thermal conductivity of the unit cell (Fig. 1a) becomes $k_{eff} = \phi_{Li} k_{Li} + (1 - \phi_{Li}) k_0$, where $R_{tot,Li}$, $R_{tot,O}$, $k_{Li}$, and $k_O$ are the thermal resistance and conductivity of the region with and without Li deposition, respectively. Note that $k_{Li}$ and $k_O$ are obtained using the calibrated thermal model (Eq. (1)) with different constriction radius due to Li plating. Therefore, the lithium coverage ($\phi_{Li}$) can be determined using the measured $k_{eff}$ and calculated $k_{Li}$ and $k_O$.

To validate the thermal model for Li plating as discussed above, an in-situ cell was built to quantify the lithium coverage using X-ray microtomography. Supplementary Fig. 4 displays the schematics of our customized polyether ether ketone (PEEK) cell holder and the components inside the cell[27,28] (see Methods for details). After three formation cycles, we charged the cell at 6 C to 50% SOC and 80% SOC with a high cutoff voltage of 4.6 V (Supplementary Fig. 5), which resulted in a sufficient amount of lithium plating. Correspondingly, we measured the thermal conductivity under the same charge conditions (Supplementary Fig. 6) and back calculated $\phi_{Li}$ using our thermal model. Note that the electrolyte consumption effect is assumed to be negligible as the experiments were finished in a few hours. Figure 2c demonstrates the good agreement between the Li coverage visualized by tomography (Fig. 2d–f) and that estimated using our approach, with a deviation of <5% (4.6% and 1.3%). This validation proves the effectiveness of our method for assessing the severe coverage of Li metal.

## Diagnostic protocol and data analysis

The calibrated model quantifies the dependence of $k_{eff}$ on the amount of electrolyte consumption ($\phi_{dry}$) and lithium coverage ($\phi_{Li}$), which can be used for battery sensing. Our diagnostic protocol consists of monitoring the evolution of $k_{eff}$ and extracting the quantitative degradation information from the measured $k_{eff}$, as summarized in Fig. 3. For nonembedded $k_{eff}$ measurement, we fabricated thermal-wave sensors on flexible polyimide films, which could be easily attached onto the surface of batteries (Fig. 1f). Details on the preparation of the sensors, thermal-conductivity measurement, and analysis can be found in Methods. The evolution of $k_{eff}$ was continuously monitored during cycling. We assumed that only one phenomenon (electrolyte

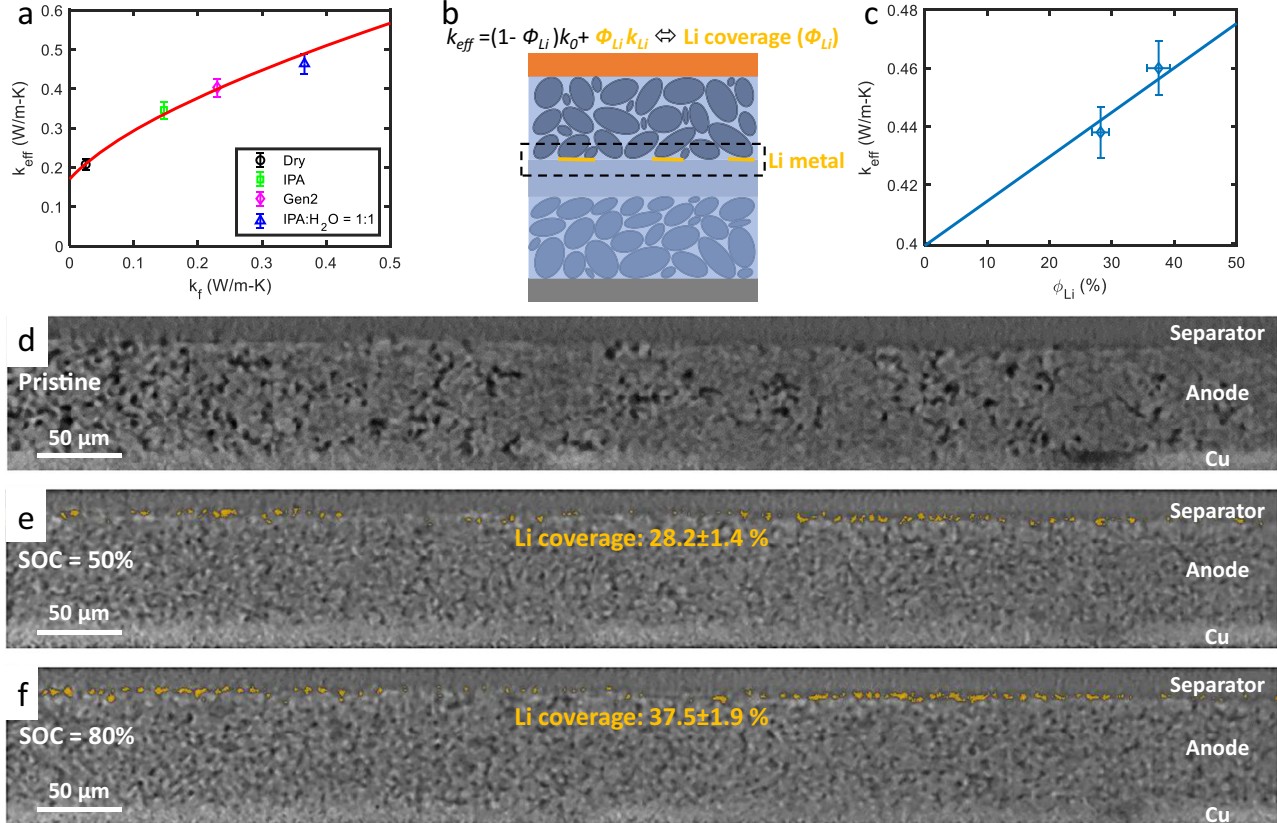

**Fig. 2 | Dependence of effective battery thermal conductivity ($k_{eff}$) on electrolyte consumption and Li coverage. a** Calibrated relationship between $k_{eff}$ and $k_f$. The accuracy of this calibrated model is verified by comparing the predicted and measured $k_{eff}$ related to different fluids. **b** Effective battery thermal conductivity due to parallel heat conduction through the regions with and without Li plating.

**c** Comparison of the predicted and measured $k_{eff}$ vs. $\phi_{Li}$ (i.e., Li coverage). The deviation is within the uncertainty range of the thermal-conductivity measurement and X-ray tomography. Representative cross-sectional slice of the graphite–separator interface in an X-ray tomogram of a cell at **d** SOC = 0%, **e** SOC = 50%, and **f** SOC = 80%.

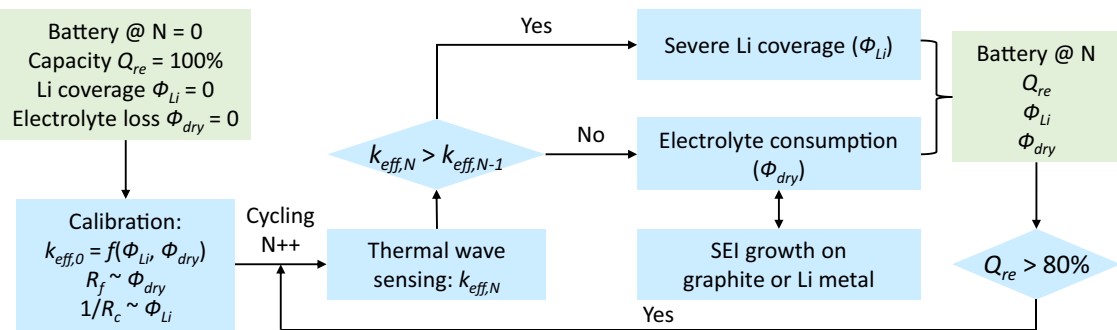

**Fig. 3 | Process flow for the measurement procedure and data analysis.** With the calibrated model, the amount of electrolyte consumption and Li coverage is back calculated from the measured $k_{eff}$. The degradation sources can be further distinguished by monitoring the electrolyte consumption rate and calibrating the rate due to SEI growth on graphite. The observed higher rate indicates the reaction between Li deposits and electrolyte (SEI growth on Li metal).

consumption or lithium coverage) dominates between two consecutive measurements, which leads to a conservative estimate of $\phi_{dry}$ and $\phi_{Li}$. Apparently, the deviation related to this assumption depends on the capacity loss ($\Delta Q$) or cycle number (N) between the measurements and can be negligible for continuous monitoring with sufficiently small $\Delta Q$ and N in between.

Figure 3 summarizes the process used to distinguish and quantify the degradation sources during cycling. The quantitative degradation information was updated after each $k_{eff}$ measurement for continuous observation until 20% capacity loss. Comparing the measured $k_{eff}$ for cycle N and N − 1 ($k_{eff,N}$ vs. $k_{eff,N-1}$), the increase of $k_{eff}$ indicates severe

lithium plating, and the coverage of Li can be quantified. Otherwise, the change is attributed to electrolyte consumption.

Further, the exact degradation source can be determined from the monitored $\Delta\phi_{dry}/\Delta Q$. As a reference, the rate of electrolyte consumption associated with the SEI growth on graphite, $(\Delta\phi_{dry}/\Delta Q)_{gr}$, is calibrated with battery operation at slow rates and early stages where lithium plating rarely occurs. Compared to the reaction on graphite, the growth of the SEI layer on Li metal is much faster due to the poor stability of the SEI and the high reactivity of Li metal[29–31]. Further, the rate of electrolyte loss due to the reaction between Li metal and the electrolyte reflects the morphology of Li deposition (e.g., dense or

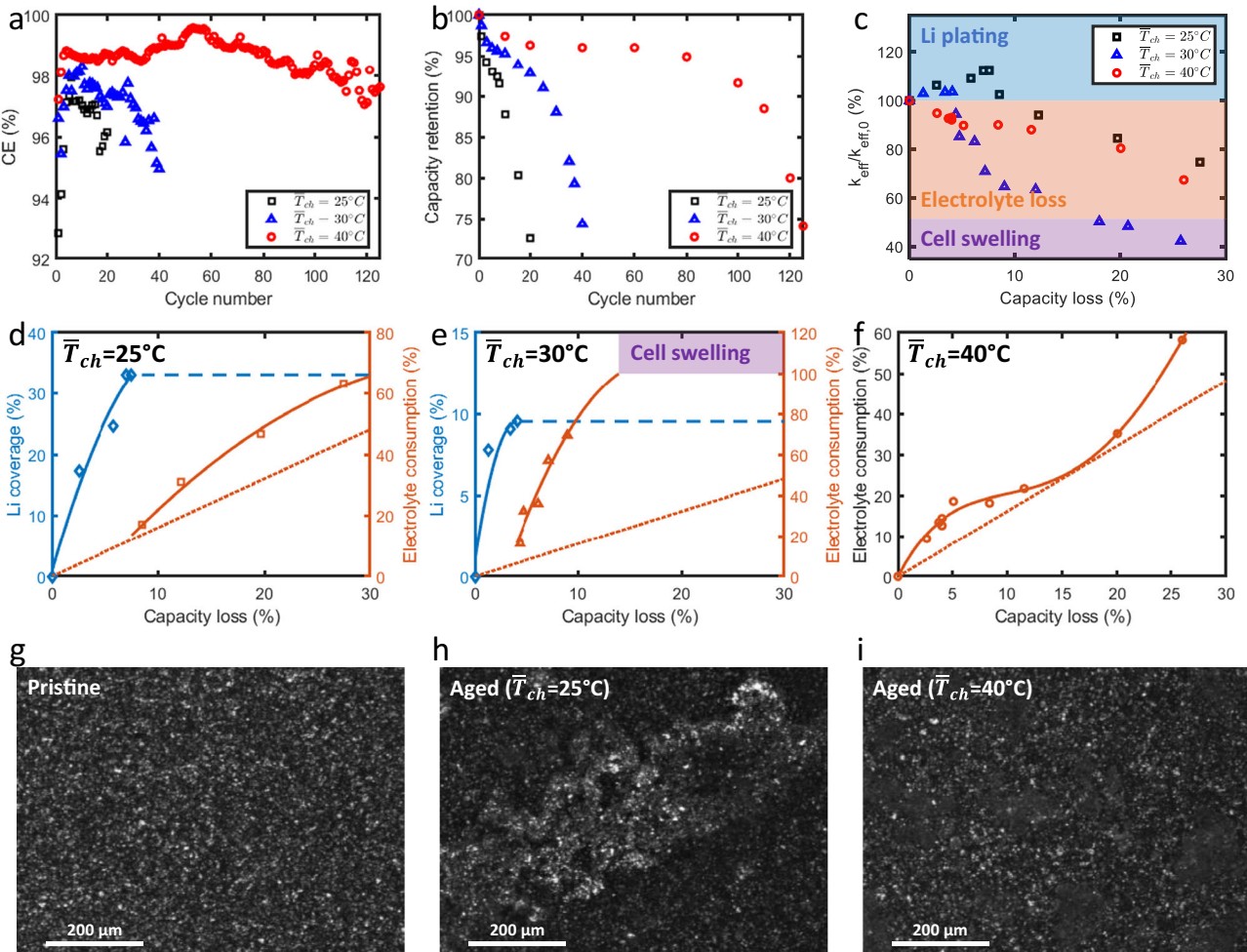

**Fig. 4 | Quantitative assessment of battery degradation during fast charging. a** Coulombic efficiency and **b** capacity retention of the cells under different thermal conditions. The high charge temperature benefits the mitigation of Li plating and leads to high CE and extended cycle life. **c** Evolution of $k_{eff}$ during cycling. The initial increase of $k_{eff}$ observed in the two low $\bar{T}_{ch}$ cases indicates the severe Li deposition and coverage, whereas this effect does not appear in the high $\bar{T}_{ch}$ case. From the measured $k_{eff}$, the evolution of degradation sources is quantified for the cases with (**d**) $\bar{T}_{ch} = 25\,°C$, (**e**) $\bar{T}_{ch} = 30\,°C$, and (**f**) $\bar{T}_{ch} = 40\,°C$. The source of electrolyte consumption can be distinguished by comparing with the consumption due to SEI growth on graphite (see the orange dotted line in panels **d**–**f**. The Li coverage is qualitatively verified with the ex situ optical images of **g** pristine and aged anodes associated with **h** $\bar{T}_{ch} = 25\,°C$ and **i** $\bar{T}_{ch} = 40\,°C$. Shiny Li metal is observed in the low $\bar{T}_{ch}$ case, whereas no severe Li coverage exists in the high $\bar{T}_{ch}$ case. The scale bar is 200 μm in panels **g**–**i**.

porous Li deposits) as the reaction rate largely depends on the surface area to volume ratios of Li deposits[32,33].

**Proof-of-concept study on fast-charging commercial LIBs**
The non-embedded nature of our attachable thermal-wave sensor is very advantageous for continuous monitoring of cycle-life battery degradation. As a proof of concept, we demonstrate that our attachable sensor can be used to quantitatively distinguish the degradation sources for fast-charging commercial LIBs. Fast charging can cause lithium plating and early battery degradation due to a complicated combination of lithium plating and side reactions. Recent studies revealed that lithium plating can be mitigated by charging at an elevated temperature[34–37]. However, the high charging temperature also accelerates the reaction with the electrolyte and may result in rapid electrolyte consumption and, hence, speed up battery degradation. Understanding and quantifying the exact degradation sources is thus critical for the design of fast-charging strategies.

The 3-Ah commercial batteries using the same NMC/Gr electrodes and electrolyte as those used in the calibration experiments were charged at 6 C to 80% SOC under various thermal conditions, leading to an average charging temperature ($\bar{T}_{ch}$) of 25 °C, 30 °C, and 40 °C,

respectively (Supplementary Fig. 7; see Methods for details on the charging protocol). For ensuring the test repeatability, 3 cells were tested for each thermal condition (Supplementary Fig. 8). Figure 4 summarizes the measurement results of representative cells. The high charging temperature resulted in high coulombic efficiency (CE) and extended cycle life compared with the lower charging temperatures (Fig. 4a, b). Figure 4c shows the variation of $k_{eff}$ with capacity loss under distinct thermal conditions. In both the low $\bar{T}_{ch}$ cases (i.e., 25 °C and 30 °C), an increase of $k_{eff}$ at the initial stage indicates different levels of lithium plating. In contrast, no severe lithium coverage was observed for high $\bar{T}_{ch}$. After this stage, the decrease of $k_{eff}$ in all cases reveals the consumption of electrolyte with capacity loss. For the two low $\bar{T}_{ch}$ cases, such a decrease does not indicate the disappearance of Li deposition. Instead of increasing the lithium coverage and $k_{eff}$, Li plating can continue in this stage and tends to appear in the region where Li deposition already exists based on the classical nucleation theory[38]. The rapid reaction between the deposited Li metal and electrolyte results in fast electrolyte consumption, which dominates the observed decrease of $k_{eff}$ (see Fig. 3 for the process flow). In addition, the measured $k_{eff}$ can be used to detect cell swelling by comparing the $k_{eff}$ with that of dry cells. When $k_{eff}$ is lower than the dry

condition, it reflects the loss of contact inside batteries due to swelling (see Supplementary Fig. 9 for the aged cell for $\bar{T}_{ch} = 30\,°C$).

Figure 4d–f summarize the quantitative assessment of degradation sources in representative cells using our approach (see Supplementary Fig. 8 for the degradation trend observed in all the cells). For the lowest $\bar{T}_{ch}$ case, the lithium coverage increases to 32.5% with 7.6% capacity loss. Compared to the calibrated $(\Delta\phi_{dry}/\Delta Q)_{gr}$ obtained at low C-rate (e.g., 1 C) for electrolyte consumption due to SEI growth on graphite (Supplementary Fig. 10), a combination of SEI growth on graphite and Li metal results in a higher electrolyte-consumption rate. Note that this rate decreases with the capacity loss due to the continuous growth of SEI and the reduction of exposed surface area for side reactions. With $\bar{T}_{ch} = 30\,°C$, the maximum Li coverage increases to 9.5%. Increasing the charging temperature benefits the mitigation of severe Li deposition. However, the effective surface area to volume ratio of Li deposits increases as the amount or volume of deposited Li decreases. The higher ratio associated with the mild lithium plating causes the faster electrolyte consumption due to the larger surface area for the reaction with the electrolyte[32,33] compared with the $\bar{T}_{ch} = 25\,°C$ case (Fig. 4d)). As a result, more electrolyte is consumed for the same capacity loss (Fig. 4e), resulting in more gas formation and the observed cell swelling (Supplementary Fig. 9). Increasing the charging temperature to 40 °C further reduces the amount of lithium plating, and no clear lithium coverage is observed (Fig. 4f). A comparison with the calibrated $(\Delta\phi_{dry}/\Delta Q)_{gr}$ reveals the consumption of electrolyte due to a small amount of lithium plating at the initial stage, which is consistent with the relatively low CE in the initial cycles (Fig. 4a). After the initial stage, the consumption rate decreases with the capacity loss and SEI growth. As a certain amount of electrolyte is consumed (e.g., ~30%), a large lithium concentration gradient is developed across the electrolyte and electrodes. This can trigger lithium plating and accelerate the electrolyte consumption due to the reaction between Li metal and the electrolyte, as shown in Fig. 4f. This interaction explains the transition of capacity fade from the linear stage to nonlinear regime (Fig. 4b).

Further, we disassembled the aged cells in an Ar-filled glovebox and measured the mass evolution until complete electrolyte evaporation. The $\phi_{dry}$ can be back calculated from the measured mass difference (see Methods), which verifies the accuracy of our method in quantifying the amount of electrolyte consumption (Supplementary Table 3). In addition, the level of lithium coverage was qualitatively verified using the images of aged anodes (Fig. 4g–i and Supplementary Fig. 11). A large portion of the aged anode associated with $\bar{T}_{ch} = 25\,°C$ is covered with shiny Li metal (Fig. 4h), whereas this effect does not exist for $\bar{T}_{ch} = 40\,°C$ (Fig. 4i). We also performed chemical titration experiments (see Methods) and relaxation voltage analysis to validate the mitigation of Li plating at high temperatures (Supplementary Table 4 and Supplementary Fig. 12). Besides, the impact of the changes in the cathode on $k_{eff}$ proved to be trivial in these studies. The variation of $k_{eff}$ associated with cathode aging is only 0.57% (Supplementary Fig. 13), which is within the uncertainty of our non-embedded thermal measurement as analyzed below. Thus, the degradation sources during fast charging at different temperatures are quantitatively distinguished using our measurement scheme, and the accuracy is verified via postmortem characterizations.

### Application of the thermal-wave sensor in various battery types and operating conditions

In this proof-of-concept study, the efficacy of our sensor and approach was evaluated using single pouch cells. We further validated the long-term stability of the sensor (Supplementary Fig. 14) and the bonding reliability across a wide temperature range (Supplementary Fig. 15). As for the detection limit of the sensor, the relative uncertainty of $k_{eff}$ is ±0.75% based on the measured relative standard deviation of thermal-wave signals (Supplementary Fig. 16). With the calibrated $k_{eff}$ vs. $\phi_{dry}$

and $k_{eff}$ vs. $\phi_{Li}$, the sensitivity threshold of our method in measuring $\phi_{dry}$ and $\phi_{Li}$ is estimated to be 1% and 2%, respectively.

Further, the sensor and methodology developed here can be applied to various battery types, e.g., prismatic and cylindrical cells. Since the measurement accuracy relies on the sensitivity to the structural change of unit cells, an essential prerequisite for this approach is that the total thermal resistance of all unit cells should dominate over that of case, i.e., $R_{uc,tot} \gg R_{case}$. In common commercial cells, this condition can be easily met in different battery formats as the total thermal resistance of all unit cells (i.e., electrodes, separators, and current collectors) is much higher than that of the case (Supplementary Table 5).

In addition, the frequency-dependent nature of thermal waves allows for controllable penetration depth (see Methods), which is advantageous for the use of our sensor in a battery stack or pack of multiple cells. By controlling the frequency ($\omega$) and penetration depth ($\propto 1/\sqrt{\omega}$), thermal waves can be localized near the sensor or extended to the bottom of the single cell or battery stack. The information of the intermediate battery can be obtained using a sensor attached to its surface, as demonstrated in our case study of the single battery. Noteworthily, we only detect the signals at precisely the frequency at which we are operating the sensor. Thus, the sensor operation is not influenced by other thermal signals or events when they are at different frequencies.

## Discussion

Understanding battery degradation typically relies on laboratory-based techniques and/or extremely limited resources at large-scale photon-based user facilities. However, battery degradation is known to be a complicated phenomenon that depends on many factors such as the electrochemical systems, temperatures, and operating conditions. The widespread application of batteries in extreme and varying conditions can cause battery degradation and safety issues that are unexpected in the laboratory. Obtaining such quantitative information using a simple and non-embedded technique is crucial for improving the safety and reliability of batteries in the real world. With the sensor and methodology developed in this work, we quantified the evolution of lithium coverage and electrolyte consumption during fast charging of commercial batteries under various thermal conditions. At low charging temperatures (e.g., 25 °C and 30 °C), sluggish kinetics induced lithium plating dominates the initial rapid capacity fade, and the reaction between the Li deposits and electrolyte further accelerates the aging process. The byproduct of this side reaction could result in cell swelling and safety issues in certain conditions (e.g., 30 °C). Charging at an elevated temperature (e.g., 40 °C) mitigates the Li plating and extends the fast-charging cycle life. Eventually, the improved performance depends on the rate of electrolyte consumption at high temperatures. As a result of the increased consumption rate, the insufficient amount of remaining electrolyte causes a large lithium concentration gradient across the electrode during fast charging, which triggers lithium plating and accelerates the capacity fade. Thus, our operando measurement provides real-time battery status as valuable feedback for battery management in various conditions. Further, the quantitative assessment of degradation sources could help to guide the design of advanced batteries, e.g., the need of optimal thermal condition and thermally stable electrolyte for fast-charging batteries. In summary, our non-embedded thermal-wave sensor enables continuous monitoring of real-world battery degradation as well as quantification of the exact degradation sources.

## Methods
### Bruggeman model
For a mixture of two materials, the Bruggeman model[23] describes the relationship between the mixture and single-phase property as $\varphi_1(\frac{k_1 - k_{mix}}{k_1 + 2k_{mix}}) + (1 - \varphi_1)(\frac{k_2 - k_{mix}}{k_2 + 2k_{mix}}) = 0$, where $\varphi_1$ is the volume fraction of

one material and $k_1$ is the corresponding thermal conductivity. $k_2$ and $k_{mix}$ are the thermal conductivity of the other material and mixture, respectively. Based on this model, the fourth parameter can be determined with the other three parameters known or measured. We use the Bruggeman model twice. 1) The model is first used to extract the thermal conductivity of the anode and cathode solid particles using the experimentally measured thermal conductivity of wet porous electrodes. In this case, $k_{mix}$ (electrode conductivity), $k_2$ (electrolyte conductivity), and $\phi_1$ are known due to the known porosity of the electrodes. 2) The model is then used to extract $k_{f,eff}$ by applying the model to experimentally obtained $k_{eff}$ from the thermal-wave sensor during various electrolyte dryout experiments. Once $k_{f,eff}$ is obtained, this model is applied again to extract the fraction of gas present in the aged cell with electrolyte dry out. In this case, $k_{mix}$ is replaced by $k_{f,eff}$, $k_1$ (i.e., the thermal conductivity of the gas) is known, and $k_2$ (i.e., the thermal conductivity of the electrolyte) is also known. Thus, $\phi$ (i.e., the volume fraction of gas) can be calculated.

## Thermal constriction resistance

We approximate the electrode particles as cylinders of radius $r_p$ and height $2r_p$. The thermal interface resistance of the solid (i.e., electrode particles and separator) consists of the thermal constriction resistance due to constriction of conduction areas and the thermal boundary resistance due to phonon mismatch. The latter term (~$10^{-8}$ m$^2$K/W[39]) is negligible compared to $R_c$, i.e., 1/1000 to 1/10 of $R_c$. To calculate the thermal contact resistance, Cooper et al. [24] proposed the simple formula as $R_c = \left(1 - \frac{a}{r_p}\right)^{1.5} / 4k_p a + \left(1 - \frac{a}{b}\right)^{1.5} / 4k_{sep} a$, where $a$, $b$, $k_p$, and $k_{sep}$ are the contact radius, lattice width, particle thermal conductivity, and separator thermal conductivity, respectively.

## Cell preparation

Supplementary Table 1 summarizes the property information (e.g., thickness, loading, and porosity) of electrodes, separator, and electrolyte used in this study. The electrode area for the sensor-embedded pouch cell is 12 cm$^2$ (3 cm × 4 cm), with embedded-sensors of 150 μm × 4.5 mm. Circular electrodes (area: 1.267 cm$^2$; diameter: 1.27 cm) were used in the in situ cell for X-ray microtomography. For the 3-Ah commercial cells, the electrode size is 5.1 cm × 10.25 cm (area: 52.275 cm$^2$) and the size of the non-embedded sensor is 300 μm × 9 mm. The volume of added electrolyte is ~1.6 and ~1.2 times that of the pore volume of the cell components for the customized single-layer cells and the 3-Ah commercial multilayer cells (thickness: 6 mm), respectively. The difference comes from the large dead volume in the customized cells compared to that in the commercial cells.

## Thermal-conductivity measurement and analysis

100-nm Cr/Au layers were deposited onto thin Kapton films (25.4 μm) using a shadow mask and CHA e-beam evaporator. The sensors were then bonded with batteries using epoxy (e.g., SU-8 used in our work) for the thermal-conductivity measurement. A Keithley 6221 AC current source was used to provide the current of frequency $\omega$ passing through the sensors and generate the temperature rise of $2\omega$ frequency ($\Delta T$). The temperature rise was determined from the corresponding voltage fluctuation of $3\omega$ frequency ($V_{3\omega}$) using an Amtek 7279 Lock-in amplifier. The measurement procedure has been detailed in prior works[40–42] and is thus not repeated. Here, we used the low-frequency slope method of $3\omega$ data analysis to determine the battery thermal conductivity, i.e., the slope $\partial \Delta T / \partial ln(\omega)$ is inversely proportional to the effective thermal conductivity ($k_{eff}$). The cross-plane battery thermal conductivity is obtained as $k = k_{eff}^2 / k_{in}$, where $k_{in}$ is the in-plane battery thermal conductivity. Note that $k_{in}$ is dominated by the high-thermal-conductivity current collector layers and the change with degradation is negligible, e.g., the decrease of $k_{in}$ is less than 0.4% as the battery is fully dried-out. Thus, we use the cross-plane thermal conductivity (i.e., $k$ in the main text) as the indicator of battery degradation.

The low frequency range of interest is estimated based on the cell thickness and thermal penetration depth $\propto 1/\sqrt{\omega}$, i.e., the penetration depth should be comparable to the cell thickness. For the 6-mm cells used in this study, the thermal conductivity was extracted using the data in the frequency range of 25 mHz to 0.5 Hz (Supplementary Fig. 16). For consistency, we collected the data when the cell was discharged to the cutoff voltage. In fact, the SOC has a negligible effect on $k$ for battery detection as it is very weak compared to the degradation effect (Supplementary Fig. 17).

## In situ cell and X-ray microtomography

NMC/Gr cells were assembled using a custom cell holder for an in situ study. The main body of the cell holder was machined from polyether ether ketone (PEEK) for the transmission of X-rays using a design adapted from Ho et al. [27] and Finegan et al. [28]. PTFE ferrule was used to keep the cell airtight. The contact between the cell and stainless-steel pins was adjusted using a hard spring. Electrode property information is summarized in Supplementary Table 1. Before exposure to the beam, three formation cycles were performed at 0.1 C in the range of 3–4.1 V. After the formation, the cell was charged to 50% and 80% SOC, and the morphology change related to Li deposition was monitored. X-ray microtomography was performed at beamline 8.3.2 at the Advanced Light Source (ALS) at Lawrence Berkeley National Laboratory. Details on the 3D reconstructions and visualizations can be found in prior works[27,28,43–45].

## Cycling experiments

Commercial 3-Ah LIBs were used in the cycling experiments. According to the manufacturer, the recommended maximum charge rate is 1 C and the nominal energy density is 180 Wh/kg. The cycle life associated with 20% capacity loss is greater than 500 times. The batteries were charged to 80% SOC using a standard constant current–constant voltage (CCCV) charge protocol. Charge rates of 1 C and 6 C were used for slow and fast charging, respectively. After a standard 10-min rest, the cell was discharged at 1 C and then at C/3 with a cutoff voltage. The recommended cutoff voltage for charging and discharging is 4.25 and 2.75 V, respectively. The rest time after discharge ranges from 15 to 30 min depending on the thermal condition and time required to reach an approximate thermal equilibrium before the next cycle. To monitor the capacity fade, the capacity is calibrated by C/3 charging and discharging after a certain number of cycles. All the cycling experiments were performed with an 8-channel Arbin Laboratory battery testing system (LBT21084), and the thermal conditions and temperature were controlled using a TestEquity thermoelectric temperature chamber (TEC1).

## Estimation of $\phi_{dry}$

This analysis relies on the volatile nature of EMC solvent in the Gen2 electrolyte, i.e., 1.2 M LiPF$_6$ in EC:EMC (3:7). We disassembled the cells in an Ar-filled glovebox and recorded the evolution of mass until complete electrolyte evaporation. The mass of evaporated EMC, i.e., remaining EMC in the cell before disassembly ($m_{EMC}$), is known from the mass difference between the initial and final states. With $m_{EMC,0}$ calibrated for the pristine cell and $m_{EMC,1}$ measured for the aged cell, the mass of consumed EMC in the aged cell is given by ($m_{EMC,0} - m_{EMC,1}$). The consumption of EMC by volume is calculated based on the ratio of EC and EMC in the electrolyte. The range of electrolyte consumption can be further estimated by assuming 0 to 100% consumption of EC, which explains the relatively large uncertainty of $\phi_{dry}$ calculated from the mass difference (Supplementary Table 3). Note that the $\phi_{dry}$ measured using our method is within this uncertainty range, with a relative deviation of 4.52% and 2.41%, respectively.

## Titration mass spectrometry

The fully discharged 3-Ah pouch cells were disassembled in an Ar-filled glovebox. The graphite electrodes were cut into 16 cm² pieces, rinsed with dimethyl carbonate (Gotion), and dried under vacuum at room temperature to remove the residual electrolyte from the surface and the pores. Then, the dried graphite electrode was placed into a custom-made three-neck glass vessel[46]. The vessel was attached to the mass spectrometry apparatus and purged with continuous Ar to remove any residual contaminants inside the vessel and the line. When the mass spectrometer stabilized, 3.5 M $H_2SO_4$ was injected through the injection port by using a gas-tight syringe (VICI, Series C syringe with Pressure-Lok side port needle). The gas was accumulated into the 2 mL sample loop and sampled every 4 min. $H_2SO_4$ reacts with dead Li and lithiated graphite ($Li_xC_6$), solid carbonates species, and lithium acetylide, and evolves $H_2$, $CO_2$, and $C_2H_2$[47,48], respectively. Evolved gases ($H_2$, $CO_2$, and $C_2H_2$) were quantified with the calibrated data based on the mass spectrometer ion current, and the total amounts of Li-containing species were determined, as summarized in Supplementary Table 4.

## Data availability

The data supporting the findings of this study are available from the corresponding author on reasonable request, and are provided in the Supplementary Code.

## Code availability

The code used in this study can be downloaded from Supplementary Code.

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

## Acknowledgements

The authors acknowledge the support received from the Energy Efficiency and Renewable Energy, Vehicle Technologies Program, of the US Department of Energy under contract no. DEAC0205CH11231 (R.P.). This work used beamline 8.3.2 at ALS, a DOE Office of Science User Facility under contract no. DEAC0205CH11231 (M.C.T.). The authors thank Dr. Dilworth Y. Parkinson for help on tomography data collection.

## Author contributions

Y.Z. and R.P. conceived the idea. Y.Z. developed the methodology and conducted the cycling experiments. F.S., B.Z., J.L., Y.Z., B.D.M. and M.C.T. performed the postmortem characterization. Y.Z., D.C., Q.Z., Y.F., S.K., S.D.L. and V.B. contributed to the sensor design and fabrication. Y.Z. and R.P. wrote the manuscript with feedback from all the authors.

## Competing interests

The authors declare no competing interests.
