## [Peer Review File · Nature Communications]

REVIEWER COMMENTS

Reviewer #1 (Remarks to the Author):

With the widespread use of lithium-ion batteries (LIBs) and the popularity of electric vehicles, the demand for monitoring real-world battery degradation is crucial for LIBs' application. The design of nonintrusive and operando sensors is the key to quantifying the LIBs' degradation. In this manuscript, a brand-new method of quantifying the evaluation of battery degradation is designed by using simple non-embedded thermal-wave sensors. In this way, the attenuation of battery capacity due to lithium deposition and electrolyte decomposition can be effectively distinguished. The designed sensor system is relatively simple, low price, and has the potential to be widely popularized. Some suggestions and questions to improve the quality and impact of this work are listed as followed:

1. The construction of the theoretical system only considers lithium plating and electrolyte consumption. The pulverization of the cathode material during the fast charging/discharging should also cause the change of interface contact and affect the cross-interface heat conduction. Please distinguish between cathode, anode, and electrolyte in the model establishment.
2. The author used optical photographs to quantify lithium metal deposits in Fig 4, which is not rigorous enough because small amounts of lithium deposits may remain in the pores of the separator and be destroyed during dismantling. As we can see in Fig S10b, the separator is darker than that in Fig S10a. The authors are requested to provide more accurate evaluation methods, such as the use of chemical titration to identify the absence of lithium metal deposits in $T_{ch}=40^{\circ}\text{C}$.
3. The authors only carried out several experiments to verify the amount of lithium deposition. The quantitative analysis of electrolyte consumption is missing. Please use experiments like gas chromatography to provide more accurate electrolyte consumption data to support the author's simulation results.
4. This paper uses a single stacked battery for testing. If multiple (>3) batteries are stacked up and down, can the accuracy of the intermediate battery test be guaranteed? Does the thermal pulse signal of the upper and lower batteries affect the intermediate battery? This paper would be much less innovative if only a single battery could be tested. In addition, the market share of laminated pouch cells is less than that of prismatic cells and cylinder cells, is the test method used in this paper universal in battery configuration?

5. As a new test method, the authors were asked to provide more than two parallel sample data to illustrate the repeatability of the test, especially in Fig 4c.

Reviewer #2 (Remarks to the Author):

This manuscript by Zeng et al. reported a non-embedded thermal-wave-sensing technology to monitor real-world batteries, and proposed a measurement scheme with the use of effective battery thermal conductivity (k_{eff}) as a quantitative indicator of battery degradation. The authors have used multiple theoretical deduction as well as structural techniques to substantiate their measurement scheme and quantitatively distinguished the amount of lithium plating and electrolyte consumption. In view of the focus on battery lifetime and safety, there is a need to develop an operando monitor technique for quantitative evaluation of battery degradation. This is especially important as acquiring quantitative degradation information is considered one of the significant challenges in operating commercial cells without additional damage. The measurement scheme present in this manuscript is a decent addition to the research on sensor technologies for batteries. However, the following points must be addressed in the revised manuscript before being accepted for publication in Nature Communications.

1. As shown in Fig. 1f and Supplementary Fig. 1a, the thermal-wave sensor is attached to the outer separator or collector, which makes contact between the sensor and the electrolyte. Thus, is it subject to electrolyte or moisture erosion? And how about the accuracy or stability of the sensor after a certain operation time? It is suggested to supplement the discussion about the operation lifetime of the sensor.
2. As an attachable sensor, does it have any requirements for the specifications of battery devices, especially the thickness and components of porous separator and electrode layers? These contents should be further discussed in conjunction with the sensitivity and detection line of the sensor.
3. The sensor is bonded with batteries using epoxy for the thermal-conductivity measurement. Can this bond be well maintained when the batteries undergo wide temperature changes?
4. To validate the thermal model, it is essential to conduct other experiment characterizations for the electrolytes' decomposition amount in batteries. This will strongly support the discussion on the function of the sensor.
5. For the reader's benefit, it is recommended to add the preparation details of various cell configurations in Methods section, especially the added amount of electrolyte. Moreover, the size (e.g., thickness, area) of the sensor and electrodes should be disclosed.
6. The conclusion section should be greatly improved to highlight the research progress. For instance, the causes and evolution process of battery degradation observed via non-embedded

thermal-wave sensors should be summarized, which might provide significant guidance for the design of advanced batteries.

Reviewer #3 (Remarks to the Author):

Comments to Author:

Recommendation: Minor revisions needed as noted.

Comments:

Manuscript ID: 438621_0

Title: Nonintrusive thermal-wave sensor for operando quantification of degradation in commercial batteries

Author(s): Yuqiang Zeng, Fengyu Shen, Buyi Zhang, Divya Chalise, Qiye Zheng, Yanbao Fu, Sumanjeet Kaur, Sean D. Lubner, Vince Battaglia, Michael C. Tucker, Ravi Prasher

The manuscript submitted by Zeng et al. has developed a nonintrusive thermal-wave sensor for quantitative, operando characterization of battery degradation in commercial cells. Specifically, their proof-of-concept battery quantitatively characterized the amount of lithium plating and electrolyte consumption associated with the side reactions of graphite anode and deposited lithium. Their approach is based on the use of effective battery thermal conductivity as a quantitative metric of battery degradation.

The paper is scientifically robust and well-written. Therefore, I recommend this paper to be published in Nature Communications after it has been improved with the following minor revisions:

- 1) I think the statement of “loss of lithium-ion inventory” and “electrolyte dry out” on lines 34 and 35 on page 2 require citations.
- 2) Please add the following citation to the “acoustic” on line 36, page 2.
 - a. Chang, W., & Steingart, D. (2021). Operando 2D acoustic characterization of lithium-ion battery spatial dynamics. *ACS Energy Letters*, 6(8), 2960-2968.
- 3) Please add the following citation to the “optical signals” on line 36, page 2.
 - a. Gao, T., Han, Y., Fraggedakis, D., Das, S., Zhou, T., Yeh, C. N., ... & Bazant, M. Z. (2021). Interplay of lithium intercalation and plating on a single graphite particle. *Joule*, 5(2), 393-414.
- 4) Line 52, the word “technique” has a typo. Fix spellings. There are a few other typos in the paper as well. Copy edit one more time.
- 5) Please cite line 52-53. “The other major concern regarding embedded sensors is their incompatibility with existing battery manufacturing ...”
- 6) On page 3, line 58, I think it would be helpful to add few types of battery degradation. For example, “parameters related to various types of battery degradation such as lithium plating, electrolyte dry out,”. Also, cite few types of degradation.
- 7) Since the authors used their proof-of-concept battery to characterize Li plating and electrolyte consumption, I think brief background on Li plating and electrolyte consumption will provide context to the interdisciplinary readership of Nature Communications as to why these two types of degradation are important to investigate. Especially during fast charging. Current introduction assumes that the reader knows about these two battery degradation processes. 2-3 sentences will suffice.
- 8) In Fig. 1. caption, specify what is “a.”
- 9) Line 83, page 4, add “The before ks ..” Do not start a sentence with a symbol.
- 10) On line 116, page 6, is C in “NMC/C” graphite? If yes, I would advise changing C to Gr and then specifying that Gr refers to graphite. Graphite anode consists of components other than carbon as well such as oxalic acid, PVDF binder, etc. Hence, calling it pure C is not correct. However, if pure carbon was used as the anode, then using C is correct. Please clarify.
- 11) On line 141, page 7, mention briefly why assuming “b” remains the same during battery aging a valid assumption.
- 12) On line 156, page 7, cite the in-situ cell design with references number 36 and 37, which are cited in methods but not in the main text.
- 13) On line 163, page 7, how close is the agreements between the quantified Li coverage using tomography and their approach. Specify numbers (percentage difference) or some sort of metric for the reader.

14) Can the authors comment on the sensitivity threshold of their developed approach? For example, they demonstrated good agreement between Li coverage using their approach and tomography when Li plating was of sufficient amount. Would this technique be able to quantify/detect the onset of Li plating or when plated Li is low (not of sufficient amount)? In tomography, usually voxel size or effective spatial resolution is used to specify the size of Li deposits that can be detected. Since the authors are comparing their technique to tomography, can they specify the size of Li deposits their developed approach can detect? Also, I think few sentences on the sensitivity of the developed approach will provide perspective to the readers.

15) On line 201, page 9, add "The before 3-Ah."

16) Line 213, page 9, cite the "classical nucleation theory."

17) Line 221, page 10, specify the low C-rate (e.g. 1C?) used.

Response Letter

We thank the Reviewers for their time and appreciate their valuable suggestions to improve the manuscript. Here we provide a detailed point-by-point response to the Reviewers' comments, and we have edited the manuscript and Supplementary Information accordingly.

Color codes used in this response letter:

Black Italic: original review comments;

Blue: our responses;

Red: revisions made in the manuscript.

Reviewer #1 (Remarks to the Author):

With the widespread use of lithium-ion batteries (LIBs) and the popularity of electric vehicles, the demand for monitoring real-world battery degradation is crucial for LIBs' application. The design of nonintrusive and operando sensors is the key to quantifying the LIBs' degradation. In this manuscript, a brand-new method of quantitating the evaluation of battery degradation is designed by using simple non-embedded thermal-wave sensors. In this way, the attenuation of battery capacity due to lithium deposition and electrolyte decomposition can be effectively distinguished. The designed sensor system is relatively simple, low price, and has the potential to be widely popularized. Some suggestions and questions to improve the quality and impact of this work are listed as followed:

Response:

We appreciate the reviewer's positive evaluation of our work, and their following constructive suggestions to improve the quality of the manuscript.

1. The construction of the theoretical system only considers lithium plating and electrolyte consumption. The pulverization of the cathode material during the fast charging/discharging should also cause the change of interface contact and affect the cross-interface heat conduction. Please distinguish between cathode, anode, and electrolyte in the model establishment.

Response:

We agree that the role of cathode change in battery thermal conductivity should be discussed. However, this effect proved to be minor in our case studies. In our experimental study of 6C1C cycling, we measured the cathode thermal conductivity ($k_{c, dry}$) and the thermal contact resistance between the cathode and separator ($TCR_{c-s, dry}$) before and after fast charging/discharging. The relative change of $k_{c, dry}$ and $TCR_{c-s, dry}$ is 6.1% and 2.4%, respectively. This results in 0.57% variation of the effective battery thermal conductivity, which is weak compared to the impact of lithium plating (~12.3%) and electrolyte consumption (~25.3%). Further, our observation agrees well with a previous *ex situ* study (*Electrochimica Acta* 250 (2017) 228–237) on the effective thermal conductivity (*i.e.*, combining both the intrinsic thermal resistance and thermal contact resistance) of pristine

and aged cathode (see Table R1). The aging induced change of cathode thermal conductivity is within the measurement uncertainty. In both studies, aging induced cathode change has a minor effect on the cathode thermal conductivity and the effective battery thermal conductivity.

For a complete discussion, we also point out that the impact of cathode change could be significant in certain extreme conditions (e.g., severe cathode pulverization), which will be studied in our future work.

Table R1. Effective thermal conductivity of pristine and aged cathode (*Electrochimica Acta* 250 (2017) 228–237)

	$k_{c,eff}$ (W/m-K)	$\Delta k_{c,eff}$ (W/m-K)
Pristine (dry)	0.17 ± 0.03	0.03
Aged (dry)	0.20 ± 0.02	
Pristine (wet)	0.56 ± 0.03	-0.02
Aged (wet)	0.54 ± 0.02	

Revision:

In the revised manuscript, we discussed the impact of cathode change on battery thermal conductivity on Line 114-118, Page 5 and Line 265-268, Page 12:

“Besides, cycling induced cathode cracking may result in loss of contact inside the cathode particles, and thus increases thermal constriction resistance and degrades the interfacial thermal transport. However, this effect proved to be weak in our case studies as discussed later. We speculate that the impact of cathode change could be significant in certain extreme conditions (e.g., severe pulverization) and should be studied in the future.”

“Besides, the impact of the changes in the cathode on k_{eff} proved to be trivial in these studies. The variation of k_{eff} associated with cathode aging is only 0.57% (Supplementary Figure 13), which is within the uncertainty of our non-embedded thermal measurement as analyzed below.”

We added Supplementary Figure 13 to the revised SI:

Supplementary Figure 13 | Variation of thermal transport properties due to cathode aging. a, schematics of thermal measurement for the aged cathode, with a sensor fabricated on the separator. b, extraction of

the thermal conductivity and thermal contact resistance by fitting to the raw data. The measured cathode thermal conductivity ($k_{c, dry}$) and thermal contact resistance ($TCR_{c-s, dry}$) in a dry condition is 0.87 W/m-K and 3.20×10^{-4} m²K/W, respectively. From our case studies, the relative variation of k_{eff} associated with cathode change is only 0.57%, which is within the uncertainty range of our non-embedded measurement and negligible compared to the effect of lithium plating and electrolyte consumption.

2. The author used optical photographs to quantify lithium metal deposits in Fig 4, which is not rigorous enough because small amounts of lithium deposits may remain in the pores of the separator and be destroyed during dismantling. As we can see in Fig S10b, the separator is darker than that in Fig S10a. The authors are requested to provide more accurate evaluation methods, such as the use of chemical titration to identify the absence of lithium metal deposits in $T_{ch}=40^{\circ}\text{C}$.

Response:

We agree with the reviewer that Li plating should be detected using more accurate methods. In the revised manuscript, we performed chemical titration experiments and relaxation voltage analysis to verify the mitigation of lithium plating at the higher temperature. Based on these characterizations, the amount of lithium deposits reduces significantly as the charging temperature increases (Supplementary Table 4 and Supplementary Figure 12).

Revision:

In the revised manuscript, we have mentioned that the verification was done using chemical titration and relaxation voltage analysis (Line 263-265, Page 12):

“We also performed chemical titration experiments (see Methods) and relaxation voltage analysis to validate the mitigation of Li plating at high temperatures (Supplementary Table 4 and Supplementary Figure 12).”

Details on the chemical titration were added to the Methods section on Line 410-422, Page 17:

“**Titration Mass Spectrometry.** The fully discharged 3-Ah pouch cells were disassembled in an Ar-filled glovebox. The graphite electrodes were cut into 16 cm² pieces, rinsed with dimethyl carbonate (Gotion), and dried under vacuum at room temperature to remove the residual electrolyte from the surface and the pores. Then, the dried graphite electrode was placed into a custom-made three-neck glass vessel⁴⁶. The vessel was attached to the mass spectrometry apparatus and purged with continuous Ar to remove any residual contaminants inside the vessel and the line. When the mass spectrometer stabilized, 3.5 M H₂SO₄ was injected through the injection port by using a gas-tight syringe (VICI, Series C syringe with Pressure-Lok side port needle). The gas was accumulated into the 2 mL sample loop and sampled every 4 min. H₂SO₄ reacts with dead Li and lithiated graphite (Li_xC₆), solid carbonates species, and lithium acetylide, and evolves H₂, CO₂, and

$C_2H_2^{47,48}$, respectively. Evolved gases (H_2 , CO_2 , and C_2H_2) were quantified with the calibrated data based on the mass spectrometer ion current, and the total amounts of Li-containing species were determined, as summarized in Supplementary Table 4.”

We added Supplementary Figure 12 to the revised SI:

Supplementary Figure 12 | Validation of mitigated lithium plating at $\bar{T}_{ch} = 40$ °C. The derivative of resting voltage after first-cycle fast charging with respect to time for a) $\bar{T}_{ch} = 25$ °C and b) $\bar{T}_{ch} = 40$ °C. The inflection point feature observed in the case of $\bar{T}_{ch} = 25$ °C confirms the severe lithium plating, while this feature disappears for $\bar{T}_{ch} = 40$ °C. c, H_2 gas evolution during titration for pristine and aged graphite anodes. For the cells discharged to 2.75V, the quantity of dead lithium and lithiated graphite in the anode can be determined from the accumulated amount of H_2 gas (Supplementary Table 4). The quantitative analysis verifies the mitigation of lithium plating during fast charging by operating at higher temperatures.

3. The authors only carried out several experiments to verify the amount of lithium deposition. The quantitative analysis of electrolyte consumption is missing. Please use experiments like gas chromatography to provide more accurate electrolyte consumption data to support the author's simulation results.

Response:

We appreciate the reviewer's comment on the verification of the amount of electrolyte consumption. For this purpose, we leveraged the volatile nature of EMC solvent in the Gen2 electrolyte, *i.e.*, 1.2 M LiPF₆ in EC:EMC (3:7). The pristine and cycled batteries were disassembled in a glove box, and the evolution of mass along with EMC evaporation was recorded until complete dry out. The consumption of EMC and electrolyte can be back calculated from the measured mass difference (Supplementary Table 3). The calculation process is detailed in the Methods section. We observed a good agreement between the amount of electrolyte consumption measured using our sensor and that estimated from the mass difference.

Revision:

In the revised manuscript, we added the verification of the amount of electrolyte consumption measured using our method on Line 256-259, Page 12:

“Further, we disassembled the aged cells in an Ar-filled glove box and measured the mass evolution until complete electrolyte evaporation. The ϕ_{dry} can be back calculated from the measured mass difference (see Methods), which agrees well with the amount of electrolyte consumption quantified using our method (Supplementary Table 3).”

We detailed the process of calculating the amount of electrolyte consumption from the mass difference in Methods section on Line 398-409, Page 17:

“**Estimation of ϕ_{dry} .** This analysis relies on the volatile nature of EMC solvent in the Gen2 electrolyte, *i.e.*, 1.2 M LiPF₆ in EC:EMC (3:7). We disassembled the cells in a glove box and recorded the evolution of mass until complete electrolyte evaporation. The mass of evaporated EMC, *i.e.*, remaining EMC in the cell before disassembly (m_{EMC}), is known from the mass difference between the initial and final states. With $m_{EMC,0}$ calibrated for the pristine cell and $m_{EMC,1}$ measured for the aged cell, the mass of consumed EMC in the aged cell is given by $(m_{EMC,0} - m_{EMC,1})$. The consumption of EMC by volume is calculated based on the ratio of EC and EMC in the electrolyte. The range of electrolyte consumption can be further estimated by assuming 0 to 100% consumption of EC, which explains the relatively large uncertainty of ϕ_{dry} calculated from the mass difference (Supplementary Table 3). Note that the ϕ_{dry} measured using our method is within this uncertainty range, with a relative deviation of 4.52% and 2.41%, respectively.”

We also added Supplementary Table 3 to the revised SI:

Supplementary Table 3. Estimation of ϕ_{dry} from the measured mass difference

	EMC evaporation (g)	EMC consumption (g)	EMC consumption (vol%)	ϕ_{dry} from mass difference (%)	ϕ_{dry} from our method (%)
Pristine cell	4.5927	-	-	-	-
Aged cell ($\bar{T}_{ch} = 25\text{ }^{\circ}\text{C}$)	1.8312	2.7615	45.28	60.13±12.35	62.85
Aged cell ($\bar{T}_{ch} = 40\text{ }^{\circ}\text{C}$)	1.9827	2.6100	42.80	56.83±12.35	58.20

4. This paper uses a single stacked battery for testing. If multiple (>3) batteries are stacked up and down, can the accuracy of the intermediate battery test be guaranteed? Does the thermal pulse signal of the upper and lower batteries affect the intermediate battery? This paper would be much less innovative if only a single battery could be tested. In addition, the market share of laminated pouch cells is less than that of prismatic cells and cylinder cells, is the test method used in this paper universal in battery configuration?

Response:

We thank the reviewer for mentioning the application of our method in various scenarios and different battery configuration. 1) the use of our sensor in a stack of multiple batteries: our sensor produces thermal waves at specific frequencies: The thermal penetration depth is $\propto 1/\sqrt{\omega}$ where ω is the frequency of AC current used for generating the periodic joule heating and thermal waves. By controlling the penetration depth via adjusting ω , thermal waves can be localized near the sensor or extended to the bottom of the battery or stack, providing spatial information for the measurements. With the controllable penetration depth, the information of the intermediate battery can be obtained using a sensor attached to its surface, as demonstrated in our case study of the single battery. 2) the effect of the thermal pulse signal of the upper and lower batteries: thanks to the frequency-dependent nature of the thermal wave signals, we only detect the signals at precisely the frequency at which we are operating the sensor. Thus, the sensor operation is not influenced by other thermal signals or events when they are at different frequencies. 3) the use of our method in other types of batteries: our thin film sensor is flexible and can be easily attached to the surface of different batteries. Since the measurement accuracy relies on the sensitivity to the structural change of unit cells, a prerequisite of our approach is that the thermal resistance of unit cells should dominate over that of case. For common commercial cells, this condition is easily satisfied as the unit cells are much thicker than the case, as summarized in Supplementary Table 5.

Revision:

In the revised manuscript, we added a new section “Application of the thermal-wave sensor in various battery types and operating conditions” on Page 13, and discussed the application of our method in various scenarios and different battery configuration (Line 279-293):

“Further, the sensor and methodology developed here can be applied to various battery types, *e.g.*, prismatic and cylindrical cells. Since the measurement accuracy relies on the sensitivity to the structural change of unit cells, an essential prerequisite for this approach is that the total thermal resistance of all unit cells should dominate over that of case, *i.e.*, $R_{uc,tot} \gg R_{case}$. In common commercial cells, this condition can be easily met in different battery formats as the total thermal resistance of all unit cells (*i.e.*, electrodes, separators, and current collectors) is much higher than that of the case (Supplementary Table 5).

In addition, the frequency-dependent nature of thermal waves allows for controllable penetration depth (see Methods), which is advantageous for the use of our sensor in a battery stack or pack of multiple cells. By controlling the frequency (ω) and penetration depth ($\propto 1/\sqrt{\omega}$), thermal waves can be localized near the sensor or extended to the bottom of the single cell or battery stack. The information of the intermediate battery can be obtained using a sensor attached to its surface, as demonstrated in our case study of the single battery. Noteworthy, we only detect the signals at precisely the frequency at which we are operating the sensor. Thus, the sensor operation is not influenced by other thermal signals or events when they are at different frequencies.”

We added Supplementary Table 5 to the revised SI:

Supplementary Table 5. Thermal resistance of unit cells and case in various battery formats

	Case		Unit cells (electrodes, separators, and current collectors)		R_{case} (m ² K/W)	$R_{uc,tot}$ (m ² K/W)	$R_{uc,tot}/R_{case}$
	Thickness (mm)	k_{case} (W/m-K)	Thickness (mm)	k_{uc} (W/m-K)			
Pouch cells 102.5×51×6 mm ³	0.11	0.39	5.78	0.40	2.80×10 ⁻⁴	0.0145	51.6
Cylindrical cells (18650)	0.25	45	7.75		5.63×10 ⁻⁶	0.0488	8662.1
Prismatic cells 173×125×45 mm ³	1.1	237	42.8		4.64×10 ⁻⁶	0.107	23053.6

5. As a new test method, the authors were asked to provide more than two parallel sample data to illustrate the repeatability of the test, especially in Fig 4c.

Response:

We appreciate the reviewer’s comment on the repeatability of the method and test. We added new data (*i.e.*, 3 cells tested for each thermal condition) to the revised version to verify the test repeatability. As summarized in Supplementary Figure 8, the degradation trend is similar in repeated measurements for each thermal condition.

Revision:

In the revised manuscript, we specified the number of cells tested for each condition on Line 215-217, Page 10, and Line 232-234, Page 11:

“For ensuring the test repeatability, 3 cells were tested for each thermal condition (Supplementary Figure 8). Fig. 4 summarizes the measurement results of representative cells.”

“Figs. 4d–f summarize the quantitative assessment of degradation sources in representative cells using our approach (see Supplementary Figure 8 for the degradation trend observed in all the cells).”

In the revised SI, we added Supplementary Figure 8 for the measurement results of all the cells:

Supplementary Figure 8 | Summary of battery degradation analysis for 9 cells. Capacity retention of the cells with a) $\bar{T}_{ch} = 25^\circ\text{C}$, c) $\bar{T}_{ch} = 30^\circ\text{C}$, and e) $\bar{T}_{ch} = 40^\circ\text{C}$. Evolution of k_{eff} during cycling for the cells under different thermal conditions, *i.e.*, b) $\bar{T}_{ch} = 25^\circ\text{C}$, d) $\bar{T}_{ch} = 30^\circ\text{C}$, and f) $\bar{T}_{ch} = 40^\circ\text{C}$. 3 cells were tested for each thermal condition for ensuring the measurement repeatability. Although the cell performance differs slightly due to the cell-to-cell variation, the cells tested in the same thermal condition degrade in a similar manner. The measurement results of cells 1, 4, and 7 were presented and discussed in the manuscript.

Reviewer #2 (Remarks to the Author):

This manuscript by Zeng et al. reported a non-embedded thermal-wave-sensing technology to monitor real-world batteries, and proposed a measurement scheme with the use of effective battery thermal conductivity (k_{eff}) as a quantitative indicator of battery degradation. The authors have used multiple theoretical deduction as well as structural techniques to substantiate their measurement scheme and quantitatively distinguished the amount of lithium plating and electrolyte consumption. In view of the focus on battery lifetime and safety, there is a need to develop an operando monitor technique for quantitative evaluation of battery degradation. This is especially important as acquiring quantitative degradation information is considered one of the significant challenges in operating commercial cells without additional damage. The measurement scheme present in this manuscript is a decent addition to the research on sensor technologies for batteries. However, the following points must be addressed in the revised manuscript before being accepted for publication in Nature Communications.

Response:

We appreciate the reviewer's positive evaluation of our manuscript. We thank the reviewer's constructive suggestions and believe that the quality of our manuscript has been greatly improved after these revisions.

1. As shown in Fig. 1f and Supplementary Fig. 1a, the thermal-wave sensor is attached to the outer separator or collector, which makes contact between the sensor and the electrolyte. Thus, is it subject to electrolyte or moisture erosion? And how about the accuracy or stability of the sensor after a certain operation time? It is suggested to supplement the discussion about the operation lifetime of the sensor.

Response:

We would like to clarify that there are two types of sensors used in this work. The sensor developed in this work (Fig. 1f) is attached onto **the outer surface of the battery**, and thus is free of electrolyte erosion. As for the moisture effect, the sensor made of Au/Cr is stable in such atmosphere for a sufficient long time. In an additional case study, the variation of electrical resistance in 6 months (Supplementary Figure 14) is comparable to the resistance change associated with the temperature fluctuation of the temperature chamber (± 1 °C), which verifies the long-term stability of our sensor.

The other sensor is on top of the current collector (Supplementary Fig. 1a in the revised SI) and is used for the **validation of our non-embedded approach**. From our previous work (*i.e.*, Ref. 22 in the manuscript), this embedded sensor is stable after 100+ cycles of battery operation, which is sufficient for this verification study (*i.e.*, 3 formation cycles and 2 fast-charging cycles).

Revision:

In the revised manuscript, we mentioned the validation of the sensor stability on Line 273-274, Page 13:

“We further validated the long-term stability of the sensor (Supplementary Figure 14) and the bonding reliability across a wide temperature range (Supplementary Figure 15).”

Added Supplementary Figure 14 to the revised Supplementary Information:

Supplementary Figure 14 | Sensor stability in 6 months. The variation of electrical resistance of our sensor is comparable to that associated with the temperature fluctuation of the temperature chamber (± 1 °C; shaded area in the plot), which verifies the sensor stability in the long term.

2. As an attachable sensor, does it have any requirements for the specifications of battery devices, especially the thickness and components of porous separator and electrode layers? These contents should be further discussed in conjunction with the sensitivity and detection line of the sensor.

Response:

We appreciate the reviewer’s comment on the requirement of using our sensor. In the revised version, we discussed this requirement and specified the sensitivity of the sensor. As the measurement accuracy relies on the sensitivity to the structural changes of unit cells, an essential prerequisite of this method is that the total thermal resistance of all unit cells (*i.e.*, electrodes, separators, and current collectors) should dominate over that of case, *i.e.*, $R_{uc,tot} \gg R_{case}$. Since the multiple unit cells are much thicker than the case in common commercial cells, this condition can be easily met in different types of cells (Supplementary Table 5).

Further, we analyzed the sensitivity of our method in detecting electrolyte consumption (ϕ_{dry}) and lithium plating (ϕ_{Li}). Based on the measured relative standard deviation (Supplementary Figure 16), the relative uncertainty of k_{eff} is $\pm 0.75\%$ which sets the detection line of the sensor. From the calibrated k_{eff} vs. ϕ_{dry} and k_{eff} vs. ϕ_{Li} , the detectable variation of ϕ_{dry} and ϕ_{Li} is 1% and 2%, respectively.

Revision:

In the revised manuscript, we discussed the requirement of using our sensor on Line 279-285, Page 13:

“Further, the sensor and methodology developed here can be applied to various battery types, *e.g.*, prismatic and cylindrical cells. Since the measurement accuracy relies on the sensitivity to the structural change of unit cells, an essential prerequisite for this approach is that the total thermal resistance of all unit cells should dominate over that of case, *i.e.*, $R_{uc,tot} \gg R_{case}$. In common commercial cells, this condition can be easily met in different battery formats as the total thermal resistance of all unit cells (*i.e.*, electrodes, separators, and current collectors) is much higher than that of the case (Supplementary Table 5).”

We also specified the sensor sensitivity on Line 274-278, Page 13:

“As for the detection limit of the sensor, the relative uncertainty of k_{eff} is $\pm 0.75\%$ based on the measured relative standard deviation of thermal-wave signals (Supplementary Figure 16). With the calibrated k_{eff} vs. ϕ_{dry} and k_{eff} vs. ϕ_{Li} , the sensitivity threshold of our method in measuring ϕ_{dry} and ϕ_{Li} is estimated to be 1% and 2%, respectively.”

Added Supplementary Table 5 to the revised SI:

Supplementary Table 5. Thermal resistance of unit cells and case in various battery formats

	Case		Unit cells (electrodes, separators, and current collectors)		R_{case} (m ² K/W)	$R_{uc,tot}$ (m ² K/W)	$R_{uc,tot}/R_{case}$
	Thickness (mm)	k_{case} (W/m-K)	Thickness (mm)	k_{uc} (W/m-K)			
Pouch cells 102.5×51×6 mm ³	0.11	0.39	5.78	0.40	2.80×10^{-4}	0.0145	51.6
Cylindrical cells (18650)	0.25	45	7.75		5.63×10^{-6}	0.0488	8662.1
Prismatic cells 173×125×45 mm ³	1.1	237	42.8		4.64×10^{-6}	0.107	23053.6

We also added Supplementary Figure 16 to the revised SI:

Supplementary Figure 16 | Low-frequency thermal signals for the k_{eff} measurement. a, representative raw $V_{3\omega}$ data from 25 mHz to 100 Hz. As the penetration depth is inversely proportional to $\sqrt{\omega}$, the bulk thermal conductivity is sensitive to the low-frequency signals. The well-developed low-frequency slope method is used to extract the k_{eff} from the thermal signals in the range of 25mHz to 0.5Hz, and a representative fit to $V_{3\omega}$ vs. ω is shown in b). c, relative standard deviation of $V_{3\omega}$. Fitting to the $V_{3\omega}$ and $\Delta V_{3\omega}$ determines the uncertainty of k_{eff} ($\pm 0.75\%$).

3. The sensor is bonded with batteries using epoxy for the thermal-conductivity measurement. Can this bond be well maintained when the batteries undergo wide temperature changes?

Response:

We thank the reviewer for mentioning the bonding reliability across a wide temperature range. In our case studies, we performed cycling tests from ~ 20 °C to ~ 45 °C as shown in Supplementary Figure 7, which verifies the bonding reliability above room temperature.

Further, we tested its reliability at a low temperature range (-20 °C to 30 °C), and no obvious crack or delamination is observed after 200 thermal cycles. Here, a thermal cycle consists of temperature drop to -20 °C and back up to 30 °C. Moreover, our thermal measurements before and after thermal cycling demonstrate negligible degradation of thermal contact between the sensor and battery (Supplementary Figure 15). Thus, the bonding is reliable across a wide temperature range.

Revision:

In the revised manuscript, we mentioned the validation of bonding reliability across a wide range of temperature on Line 273-274, Page 13:

“We further validated the long-term stability of the sensor (Supplementary Figure 14) and the bonding reliability across a wide temperature range (Supplementary Figure 15).”

We added Supplementary Figure 15 to the revised SI:

Supplementary Figure 15 | Sensor bonding reliability across a wide range of temperatures. a, temperature profile of a thermal cycle between 30 °C and -20 °C. b, thermal-wave signals collected before and after 200 thermal cycles. Optical images of the sensor c) before and d) after thermal cycling. No degradation of bonding is observed from these characterizations.

4. To validate the thermal model, it is essential to conduct other experiment characterizations for the electrolytes' decomposition amount in batteries. This will strongly support the discussion on the function of the sensor.

Response:

We agree with the reviewer that the amount of electrolyte consumption should be characterized using other methods for validating the thermal model and case studies. For this purpose, we leveraged the volatile nature of EMC solvent in the Gen2 electrolyte, *i.e.*, 1.2 M LiPF₆ in EC:EMC (3:7). The pristine and cycled batteries were disassembled in a glove box, and the evolution of mass along with EMC evaporation was recorded until complete dry out. The consumption of EMC and electrolyte can be back calculated from the measured mass difference (Supplementary Table 3). The calculation process is detailed

in the Methods section. We observed a good agreement between the amount of electrolyte consumption measured using our sensor and that estimated from the mass difference, which verifies the accuracy of our method in quantifying the electrolyte consumption.

Revision:

In the revised manuscript, we added the verification of the amount of electrolyte consumption on Line 256-259, Page 12:

“Further, we disassembled the aged cells in an Ar-filled glovebox and measured the mass evolution until complete electrolyte evaporation. The ϕ_{dry} can be back calculated from the measured mass difference (see Methods), which verifies the accuracy of our method in quantifying the amount of electrolyte consumption (Supplementary Table 3).”

We provided the process of calculating the amount of electrolyte consumption from the mass difference in the Methods section on Line 398-409, Page 17:

“**Estimation of ϕ_{dry} .** This analysis relies on the volatile nature of EMC solvent in the Gen2 electrolyte, *i.e.*, 1.2 M LiPF₆ in EC:EMC (3:7). We disassembled the cells in a glove box and recorded the evolution of mass until complete electrolyte evaporation. The mass of evaporated EMC, *i.e.*, remaining EMC in the cell before disassembly (m_{EMC}), is known from the mass difference between the initial and final states. With $m_{EMC,0}$ calibrated for the pristine cell and $m_{EMC,1}$ measured for the aged cell, the mass of consumed EMC in the aged cell is given by ($m_{EMC,0} - m_{EMC,1}$). The consumption of EMC by volume is calculated based on the ratio of EC and EMC in the electrolyte. The range of electrolyte consumption can be further estimated by assuming 0 to 100% consumption of EC, which explains the relatively large uncertainty of ϕ_{dry} from the mass difference. Note that the ϕ_{dry} measured using our method is within this uncertainty range, with a relative deviation of 4.52% and 2.41%, respectively.”

We added Supplementary Table 3 to the revised SI:

Supplementary Table 3. Estimation of ϕ_{dry} from the measured mass difference

	EMC evaporation (g)	EMC consumption (g)	EMC consumption (vol%)	ϕ_{dry} from mass difference (%)	ϕ_{dry} from our method (%)
Pristine cell	4.5927	-	-	-	-
Aged cell ($\bar{T}_{ch} = 25\text{ }^{\circ}\text{C}$)	1.8312	2.7615	45.28	60.13±12.35	62.85
Aged cell ($\bar{T}_{ch} = 40\text{ }^{\circ}\text{C}$)	1.9827	2.6100	42.80	56.83±12.35	58.20

5. For the reader's benefit, it is recommended to add the preparation details of various cell configurations in Methods section, especially the added amount of electrolyte. Moreover, the size (*e.g.*, thickness, area) of the sensor and electrodes should be disclosed.

Response:

We appreciate the reviewer’s comment on the preparation details of various cells used in our study. In the revised version, we added this information and specified the size of the sensor and electrodes.

Revision:

In the revised manuscript, we added “Cell preparation” in Methods section on Line 342-351, Page 15:

“**Cell preparation.** Supplementary Table 1 summarizes the property information (*e.g.*, thickness, loading, and porosity) of electrodes, separator, and electrolyte used in this study. The electrode area for the sensor-embedded pouch cell is 12 cm² (3 cm × 4 cm), with embedded-sensors of 150 μm × 4.5 mm. Circular electrodes (area: 1.267 cm²; diameter: 1.27 cm) were used in the *in situ* cell for X-ray microtomography. For the 3-Ah commercial cells, the electrode size is 5.1 cm × 10.25 cm (area: 52.275 cm²) and the size of the non-embedded sensor is 300 μm × 9 mm. The volume of added electrolyte is ~1.6 and ~1.2 times that of the pore volume of the cell components for the customized single-layer cells and the 3-Ah commercial multilayer cells (thickness: 6 mm), respectively. The difference comes from the large dead volume in the customized cells compared to that in the commercial cells.”

In the SI, the property information is summarized in Supplementary Table 1:

Supplementary Table 1. Materials & Properties

Graphite anode	Cu foil thickness: 10 μm Coating thickness: 70 μm Bulk porosity: 37.4% Coating loading: 9.38 mg/cm ² Coating density: 1.34 g/cm ³ Estimated areal capacity: 2.84 mAh/cm ² 91.83 wt% Superior Graphite SLC 1506T 2 wt% Timcal C45 carbon 6 wt% Kureha 9300 PVDF binder 0.17 wt% Oxalic acid
NMC cathode	Al foil thickness: 20 μm Coating thickness: 71 μm Bulk porosity: 33.1% Coating loading: 18.57 mg/cm ² Coating density: 2.62 g/cm ³ Estimated areal capacity: 2.67 mAh/cm ² 90 wt% Toda NMC532

	5 wt% Timcal C45 carbon 5 wt% Solvay 5130 PVDF binder
Separator (Celgard 2400)	Thickness: 25 μm Bulk porosity: 41%
Gen 2 Electrolyte	1.2 M LiPF ₆ in EC:EMC (3:7)

6. The conclusion section should be greatly improved to highlight the research progress. For instance, the causes and evolution process of battery degradation observed via non-embedded thermal-wave sensors should be summarized, which might provide significant guidance for the design of advanced batteries.

Response & Revision:

We agree with the reviewer that the conclusion section should be improved. In the revised manuscript, we summarized the causes and evolution of battery degradation observed via our sensors, and discussed the guidance for the design of advanced batteries (Line 302-316, Page 14):

“With the sensor and methodology developed in this work, we quantified the evolution of lithium coverage and electrolyte consumption during fast charging of commercial batteries under various thermal conditions. At low charging temperatures (*e.g.*, 25 °C and 30 °C), sluggish kinetics induced lithium plating dominates the initial rapid capacity fade, and the reaction between the Li deposits and electrolyte further accelerates the aging process. The byproduct of this side reaction could result in cell swelling and safety issues under certain conditions (*e.g.*, 30 °C). Charging at an elevated temperature (*e.g.*, 40 °C) mitigates the Li plating and extends the fast-charging cycle life. Eventually, the improved performance depends on the rate of electrolyte consumption at high temperatures. As a result of the increased consumption rate, the insufficient amount of remaining electrolyte causes a large lithium concentration gradient across the electrode during fast charging, which triggers lithium plating and accelerates the capacity fade. Thus, our operando measurement provides real-time battery status as valuable feedback for battery management in various conditions. Further, the quantitative assessment of degradation sources could help to guide the design of advanced batteries, *e.g.*, the need of optimal thermal condition and thermally stable electrolyte for fast-charging batteries.”

Reviewer #3 (Remarks to the Author):

Comments to Author:

Recommendation: Minor revisions needed as noted.

Comments:

Manuscript ID: 438621_0

Title: Nonintrusive thermal-wave sensor for operando quantification of degradation in commercial batteries

Author(s): Yuqiang Zeng, Fengyu Shen, Buyi Zhang, Divya Chalise, Qiye Zheng, Yanbao Fu, Sumanjeet Kaur, Sean D. Lubner, Vince Battaglia, Michael C. Tucker, Ravi Prasher

The manuscript submitted by Zeng et al. has developed a nonintrusive thermal-wave sensor for quantitative, operando characterization of battery degradation in commercial cells. Specifically, their proof-of-concept battery quantitatively characterized the amount of lithium plating and electrolyte consumption associated with the side reactions of graphite anode and deposited lithium. Their approach is based on the use of effective battery thermal conductivity as a quantitative metric of battery degradation.

The paper is scientifically robust and well-written. Therefore, I recommend this paper to be published in Nature Communications after it has been improved with the following minor revisions:

Response:

We thank the reviewer for the positive evaluation and helpful suggestions. We made all the revisions and believe that the quality of our manuscript has been greatly improved after these revisions.

1) I think the statement of “loss of lithium-ion inventory” and “electrolyte dry out” on lines 34 and 35 on page 2 require citations.

Response & Revision:

Added Ref. 6 for “loss of lithium-ion inventory” and Ref. 7 for “electrolyte dry out” in the revised manuscript on Line 63-64, Page 3 in the revised manuscript.

6. Birkel, C. R., Roberts, M. R., McTurk, E., Bruce, P. G. & Howey, D. A. Degradation diagnostics for lithium ion cells. *J. Power Sources* **341**, 373–386 (2017).

7. Palacín, M. R. & De Guibert, A. Batteries: Why do batteries fail? *Science* (80-.). **351**, (2016).

2) Please add the following citation to the “acoustic” on line 36, page 2.

a. Chang, W., & Steingart, D. (2021). Operando 2D acoustic characterization of lithium-ion battery spatial dynamics. *ACS Energy Letters*, 6(8), 2960-2968.

Response & Revision:

Added: Ref. 16 on Line 41, Page 2 in the revised manuscript.

16. Chang, W. & Steingart, D. Operando 2D Acoustic Characterization of Lithium-Ion Battery Spatial Dynamics. *ACS Energy Lett.* **6**, 2960–2968 (2021).

3) Please add the following citation to the “optical signals” on line 36, page 2.

a. Gao, T., Han, Y., Fraggedakis, D., Das, S., Zhou, T., Yeh, C. N., ... & Bazant, M. Z. (2021). Interplay of lithium intercalation and plating on a single graphite particle. *Joule*, 5(2), 393-414.

Response & Revision:

Added: Ref. 20 on Line 42, Page 2 in the revised manuscript.

20. Gao, T. *et al.* Interplay of Lithium Intercalation and Plating on a Single Graphite Particle. *Joule* **5**, 393–414 (2021).

4) Line 52, the word “technique” has a typo. Fix spellings. There are a few other typos in the paper as well. Copy edit one more time.

Response & Revision:

Corrected and used “**technique**”. We also checked the spelling throughout the manuscript.

5) Please cite line 52-53. “The other major concern regarding embedded sensors is their incompatibility with existing battery manufacturing ...”

Response & Revision:

Added Refs. 4 and 21 to this sentence (Line 58, Page 2):

4. Zeng, Y., Chalise, D., Lubner, S. D., Kaur, S. & Prasher, R. S. A review of thermal physics and management inside lithium-ion batteries for high energy density and fast charging. *Energy Storage Mater.* **41**, 264–288 (2021).

21. Huang, J., Boles, S. T. & Tarascon, J.-M. Sensing as the key to battery lifetime and sustainability. *Nat. Sustain.* **5**, 194–204 (2022).

6) On page 3, line 58, I think it would be helpful to add few types of battery degradation. For example, “parameters related to various types of battery degradation such as lithium plating, electrolyte dry out,”. Also, cite few types of degradation.

Response & Revision:

Added a few types of degradation on Line 63-64, Page 3 in the revised manuscript: “parameters related to various types of battery degradation^{6,7} such as lithium plating, electrolyte dry out, and loss of active material”.

7) *Since the authors used their proof-of-concept battery to characterize Li plating and electrolyte consumption, I think brief background on Li plating and electrolyte consumption will provide context to the interdisciplinary readership of Nature Communications as to why these two types of degradation are important to investigate. Especially during fast charging. Current introduction assumes that the reader knows about these two battery degradation processes. 2-3 sentences will suffice.*

Response & Revision:

Agreed and highlighted the importance of Li plating and electrolyte consumption in the revised version (Line 38-40, Page 2): “For example, lithium plating dominates the capacity fade during battery operation at high rates and/or low temperatures, while high operation temperature accelerates the growth of SEI and the consumption of electrolyte, leading to rapid capacity loss.”

8) *In Fig. 1. caption, specify what is “a.”*

Response & Revision:

Specified “a” as “a (*i.e.*, the constriction radius)”.

9) *Line 83, page 4, add “The before ks ..” Do not start a sentence with a symbol.*

Response & Revision:

Added “The” on Line 88, Page 4 in the revised manuscript.

10) *On line 116, page 6, is C in “NMC/C” graphite? If yes, I would advise changing C to Gr and then specifying that Gr refers to graphite. Graphite anode consists of components other than carbon as well such as oxalic acid, PVDF binder, etc. Hence, calling it pure C is not correct. However, if pure carbon was used as the anode, then using C is correct. Please clarify.*

Response & Revision:

Used “Gr” in the revised version (Line 125, 212, and 374).

11) *On line 141, page 7, mention briefly why assuming “b” remains the same during battery aging a valid assumption.*

Response & Revision:

Provided a brief explanation on Line 150-151, Page 7: “we assume that *b* will remain the same as the variation of electrode area with aging is negligible”.

12) *On line 156, page 7, cite the in-situ cell design with references number 36 and 37, which are cited in methods but not in the main text.*

Response & Revision:

Added these two references on Line 167, Page 8 and updated the reference number: “Supplementary Figure 4 displays the schematics of our customized polyether ether ketone (PEEK) cell holder and the components inside the cell^{27,28} (see Methods for details).”

13) On line 163, page 7, how close is the agreements between the quantified Li coverage using tomography and their approach. Specify numbers (percentage difference) or some sort of metric for the reader.

Response & Revision:

Specified the deviation on Line 174, Page 8: “with a deviation of < 5% (4.6% and 1.3%)”.

14) Can the authors comment on the sensitivity threshold of their developed approach? For example, they demonstrated good agreement between Li coverage using their approach and tomography when Li plating was of sufficient amount. Would this technique be able to quantify/detect the onset of Li plating or when plated Li is low (not of sufficient amount)? In tomography, usually voxel size or effective spatial resolution is used to specific the size of Li deposits that can be detected. Since the authors are comparing their technique to tomography, can they specify the size of Li deposits their developed approach can detect? Also, I think few sentences on the sensitivity of the developed approach will provide perspective to the readers.

Response:

We appreciate the reviewer’s comment on the sensitivity of our method. In the revised version, we evaluated the sensitivity of our approach and specified the minimum amount of plated Li that can be measured using our technique.

Revision:

In the revised manuscript, we discussed the sensitivity threshold of our method and specified the minimum ϕ_{dry} and ϕ_{Li} that can be measured (Line 274-278, Page 13):

“As for the detection limit of the sensor, the relative uncertainty of k_{eff} is $\pm 0.75\%$ based on the measured relative standard deviation of thermal-wave signals (Supplementary Figure 16). With the calibrated k_{eff} vs. ϕ_{dry} and k_{eff} vs. ϕ_{Li} , the sensitivity threshold of our method in measuring ϕ_{dry} and ϕ_{Li} is estimated to be 1% and 2%, respectively.”

Added Supplementary Figure 16 to the revised SI:

Supplementary Figure 16 | Low-frequency thermal signals for the k_{eff} measurement. a, representative raw $V_{3\omega}$ data from 25 mHz to 100 Hz. As the penetration depth is inversely proportional to $\sqrt{\omega}$, the bulk thermal conductivity is sensitive to the low-frequency signals. The well-developed low-frequency slope method is used to extract the k_{eff} from the thermal signals in the range of 25mHz to 0.5Hz, and a representative fit to $V_{3\omega}$ vs. ω is shown in b). c, relative standard deviation of $V_{3\omega}$. Fitting to the $V_{3\omega}$ and $\Delta V_{3\omega}$ determines the uncertainty of k_{eff} ($\pm 0.75\%$).

15) On line 201, page 9, add “The before 3-Ah.”

Response & Revision:

Added “The” before “3-Ah” on Line 212, Page 10 in the revised manuscript.

16) Line 213, page 9, cite the “classical nucleation theory.”

Response & Revision:

Added Ref. 38 for the classical nucleation theory on Line 226, Page 10 in the revised manuscript.

38. Kalikmanov, V. I. Classical Nucleation Theory. in 17–41 (2013). doi:10.1007/978-90-481-3643-8_3.

17) Line 221, page 10, specify the low C-rate (e.g. 1C?) used.

Response & Revision:

Specified the low C-rate on Line 235, Page 11 in the revised manuscript: (*e.g.*, 1C).

REVIEWERS' COMMENTS

Reviewer #1 (Remarks to the Author):

The authors have addressed the reviewer's concerns. This work is recommended to be published with current form.

Reviewer #2 (Remarks to the Author):

In view of the extensive revision and additional experiments carried out by the authors, the revised manuscript has improved significantly to merit publication in the journal Nature Communications.